# Positive Effects of UV-Photofunctionalization of Titanium Oxide Surfaces on the Survival and Differentiation of Osteogenic Precursor Cells—An In Vitro Study

**DOI:** 10.3390/jfb13040265

**Published:** 2022-11-25

**Authors:** Marco Roy, Alessandro Corti, Barbara Dorocka-Bobkowska, Alfonso Pompella

**Affiliations:** 1Department of Prosthodontics and Gerostomatology, Poznan University of Medical Sciences, 60-792 Poznan, Poland; 2Department of Translational Research and New Technologies in Medicine and Surgery, University of Pisa Medical School, 56126 Pisa, Italy

**Keywords:** titanium oxide, UV-photofunctionalization, implant osteointegration, AFM, Runx2, ALP

## Abstract

Introduction: The UVC-irradiation (“UV-photofunctionalization”) of titanium dental implants has proved to be capable of removing carbon contamination and restoring the ability of titanium surfaces to attract cells involved in the process of osteointegration, thus significantly enhancing the biocompatibility of implants and favoring the post-operative healing process. To what extent the effect of UVC irradiation is dependent on the type or the topography of titanium used, is still not sufficiently established. Objective: The present study was aimed at analyzing the effects of UV-photofunctionalization on the TiO_2_ topography, as well as on the gene expression patterns and the biological activity of osteogenic cells, i.e., osteogenic precursors cultured in vitro in the presence of different titanium specimens. Methodology: The analysis of the surface roughness was performed by atomic force microscopy (AFM) on machined surface grade 2, and sand-blasted/acid-etched surface grades 2 and 4 titanium specimens. The expression of the genes related with the process of healing and osteogenesis was studied in the MC3T3-E1 pre-osteoblastic murine cells, as well as in MSC murine stem cells, before and after exposure to differently treated TiO_2_ surfaces. Results: The AFM determinations showed that the surface topographies of titanium after the sand-blasting and acid-etching procedures, look very similar, independently of the grade of titanium. The UVC-irradiation of the TiO_2_ surface was found to induce an increase in the cell survival, attachment and proliferation, which was positively correlated with an increased expression of the osteogenesis-related genes Runx2 and alkaline phosphatase (ALP). Conclusion: Overall, our findings expand and further support the current view that UV-photofunctionalization can indeed restore biocompatibility and osteointegration of TiO_2_ implants, and suggest that this at least in part occurs through a stimulation of the osteogenic differentiation of the precursor cells.

## 1. Introduction

Dental implantology, a field of dentistry, has become a standard in dental treatments to restore the lost function and aesthetics in edentulous or partially edentulous patients [1]. Dental implants aim to simulate the root-crown apparatus in the most physiological manner, as it is inserted into the root-bearing parts of the mandible or maxilla with a prosthetic restoration on top, either screw retained or cemented. However, current outcomes show that there is a need to improve treatments, based on dental implants with respect to healing time, ageing and anatomical limitations. According to Lee et al. [2] the survival rate for an implant today is around 92% over a period of 5 years, while Norowski et al. [3] reported it to be around 89% over a period of 10–15 years, though the dental infection risk may be as high as 14%. A lot of effort has been made to improve the chemical and topographical aspects of titanium, in order to enhance the biological principles underlying osteointegration [4]. Different chemical and physical approaches (abrasion, anodization, acid-etching, plasma spraying) have been used, in an effort to improve the surface properties of the implant materials. To guarantee the stability and long life of an implant, good bone anchorage needs to be achieved, or in other words, its stability is dependent on the so-called bone-implant-contact (BIC). Nevertheless, the BIC range value is generally between 45 ± 16%, far below the ideal 100% mark. To increase the long term success rate, it is imperative to enhance the integration between the biocompatible materials and soft and hard tissues. Ideally, an increased activity should be obtained of cells capable of accelerating the process of healing and osteointegration.

It has been documented that titanium surfaces constantly attract organic impurities, such as polycarbonyls and hydrocarbons from the atmosphere, water, and cleaning solutions used during the final decontamination of implants before packaging [5,6,7,8]. Such contamination with hydrocarbons, known as biological aging, can be regarded as a physiological phenomenon, resulting in an increase of carbon levels at the implant surface from 20% (on freshly produced titanium) up to approx. 60–75% (4 weeks after production) [1,6,9,10,11,12,13]. The adsorption of hydrocarbons makes the titanium surface hydrophobic, and can actually create a coat around the surface causing an insufficient attraction of stem/progenitor cells involved in the healing process, thus hindering the complete osteointegration of implants. A machined surface grade 2, a sand-blasted/acid-etched surface grade 2, and a sand-blasted/acid-etched surface grade 4 titanium were studied.

A recently introduced procedure, termed UV-photofunctionalization, has attracted considerable attention and interest, as it is reported as a method for modifying titanium surfaces and restoring their biological compatibility, thus reversing the effects of biological ageing. Studies performed on murine pre-osteoblasts (MC3T3 cell line) demonstrated an increased protein adsorption, an improved cell attachment and proliferation, as well as an enhanced osteoblastic differentiation after the UVC irradiation of titanium surfaces [14]. However, the differences between the treated and non-treated surfaces was evaluated at short time intervals, usually no longer than 48 h. The enhancement in the osteoblastic adhesion and growth in such a small time window is anyway postulated to improve the implant’s outcome. Protein adsorption has been reported to be 80 to 300% increased after UV-photofunctionalization, as compared to non-irradiated surfaces, and the activity of osteoblasts (evaluated through specific markers, such as the ALP expression) was also significantly increased [5,7,15,16,17]. Similarly, other studies reported an overall accelerated and stronger cell adhesion to the UVC-irradiated surfaces [14,18].

The question remains however still open, whether the UVC irradiation is able to produce a carbon free surface independent of the type or topography of titanium. In order to achieve a better understanding of the osteointegration processes following the photofunctionalization, the present study was aimed at analyzing the effects of UVC irradiation on the topography of TiO_2_ used for the production of implants, as well as on the biological activity and gene expression patterns of specific osteogenic cell types, using in vitro cultures of osteogenic progenitors. The results obtained further substantiate the potential role of UV-photofunctionalization as an effective tool, in order to enhance the osteointegration and stability of titanium implants, thus prolonging their functional life.

## 2. Materials and Methods

### 2.1. Titanium Specimens

The cells were cultured on disk-shaped, commercially pure titanium specimens (10 mm × 2 mm). The specimens were divided into three groups depending on three different surface treatments: (a) a machined surface titanium grade 2, (b) a sand-blasted/acid-etched surface grade 2, and (c) a sand-blasted/acid-etched surface grade 4. Surfaces (b) and (c) were obtained with a blasting procedure using aluminum oxide particles, followed by an acid-etching procedure with hydrofluoric acid at room temperature [19], followed by a further etching step with sulfuric acid. Then all disks were rinsed with distilled water and cleaned in an ultra-sound machine.

### 2.2. UVC Apparatus for Photofunctionalization

The UVC light was delivered to specimens using a Therabeam Superosseo apparatus (Ushio Inc., Tokyo, Japan), with cycles of 12 min. The implants and discs were placed on a dedicated tray, in order to obtain the optimal and uniform irradiation.

### 2.3. Atomic Force Microscopy (AFM) Analysis

Topography of the titanium discs was analysed by AFM. The study was performed with a Solver P47 NT-MDT instrument worked in a non-contact mode. Areas of analysis on specimens was set to 50 µm × 50 µm, as well as to about 20 µm × 20 µm, in order to verify whether and to what extent the measured parameters were possibly affected when the surfaces of the different areas were analyzed on the same specimen. For comparison, a commercial TiO_2_ implant was also processed (Rapid, Osteoplant Co., Poznan, Poland), on which a smaller 15 µm × 15 µm area was measured, considering the convex implant surface. Prior to the analysis, the specimens were cleaned with isopropanol in an ultrasound washer.

### 2.4. Osteoblastic Differentiation Studies

The differentiation studies were performed using both murine osteoblast-like MC3T3-E1 cells, and murine mesenchymal stem cells.

#### 2.4.1. Studies with the Pre-Osteoblastic MC3T3-E1 Cells

Stock MC3T3-E1 cells were cultured in an undifferentiated state in DMEM supplemented with 10% fetal bovine serum (FBS, EuroClone, Pero, Italy), 100 U/mL penicillin and 100 U/mL streptomycin. At a 70–80% confluence, the cells were trypsinized and plated for expansion. The osteogenic differentiation was induced by supplementing DMEM with 2% FBS, 50 µg/mL ascorbate-2-phosphate, 10^−7^ M dexamethasone and 10 mM β-glycerophosphate (Sigma-Aldrich, St. Louis, MI, USA). The cells were maintained at 37 °C in a humidified 5% CO_2_ environment, and the culture media were replaced every three days.

#### 2.4.2. Studies with Murine Primary Mesenchymal Stem Cells (MSCs)

The mice were sacrificed by cervical dislocation. The bone marrow was obtained from the tibias and femurs and the cells were seeded using MesenCult basal medium, supplemented with 20% Mesenchymal Mouse Stimulatory Supplement and 1% Pen-Strept complete medium (Life Technologies, Monza, Italy). The cells were grown at 37 °C in a humidified atmosphere at 5% CO_2_, trypsinized at confluence and reseeded at 2 × 10^4^ cells/cm^2^ (passage 1, p1). All experiments were performed at passage 2 (p2). The cellular density seeded onto the disks was 10^5^/cm^2^. The cell count was performed at the undifferentiated state and after differentiation at 12, 24, 48 h and 8 days. The RT-PCR was performed at 0, 3 and 8 days. The viable and dead cells were evaluated with Trypan blue exclusion. All experimental protocols on mice were conducted in compliance with the Italian DL 26/2014 act, the implementation of the European Directive 2010/63 on the protection of animals used for scientific purposes. All experimental protocols were approved by the Institutional Ethics Committee for Animal Use of the University of Pisa.

#### 2.4.3. RNA Processing

The reverse transcription to cDNA was performed directly from cultured cell lysate using the TaqMAN Gene Expression Cells-to-Ct Kit (Ambion), following the manufacturer’s instructions. Briefly, the cultured cells were lysed with lysis buffer and cell lysates were reverse transcribed using the RT Enzyme Mix and appropriate RT buffer. Finally, the cDNA was amplified by real-time PCR using the Taq-Man Gene Expression Master Mix and the corresponding gene-specific assays. The gene expression levels were normalized to the expression of the housekeeping gene RPL13A and expressed as fold changes relative to the untreated mMSCs. The delta/delta calculation method [20] was used for quantification. Forward and reverse primers and probes for the selected genes were designed using primer express software (Applied Biosystems, Monza, Italy) and are listed in Table 1.

All PCR reactions were performed in a 20 µL volume using the ABI PRISM 7500. The reactions contained 10 µL 29 Taq-Man universal PCR master mix (Applied Biosystems), 400 nM of each primer and 200 nM of the probe, and cDNA. The amplification profile was initiated by 10-min incubation at 95 °C, followed by the two-step amplification of 15 s at 95 °C and 60 s at 60 °C for 40 cycles. The non-template controls were included in all experiments to exclude the reagent’s contamination. The PCRs were performed with two biological replicates.

#### 2.4.4. Statistics

All results are expressed as mean. The differences between the experimental groups (UV-photofunctionalized discs and non-treated discs) were evaluated by Student’s *t*-test. A value of *p* < 0.05 was considered statistically significant.

## 3. Results

### 3.1. AFM Analysis for the Determination of Roughness

Atomic force microscopy is an invaluable technique to measure small samples with a great degree of accuracy. It is imperative to use this kind of analysis to verify that the UVC treatment is not changing the topography of the surface after the company manufacturing.

The surface roughness plays a significant role in the cell behaviour during the process of osteointegration. Using XPS and AES analyses we previously demonstrated that the surface composition of the discs is the same as the one of the dental implants [21], however no information was obtained regarding the surface roughness. Therefore, the AFM analysis was performed in order to verify if the discs used for the biological studies have the same surface roughness as the dental implants.

The analysis was performed in a non-contact mode, in which the cantilever vibrates under the surface with a fixed frequency. The topographic images result from the measurements of the offset from the resonance frequency of the cantilever during the effect with the surface. Figure 1 shows the topographic images of the machined titanium discs, for the sand-blasted/acid-etched grade 2 and grade 4 titanium, respectively. Prior to the analysis, the samples were cleaned in an isopropanol medium in an ultrasonic washer. The analysis was performed in areas with two different sizes, 50 µm × 50 µm and 20 µm × 20 µm, in order to evaluate how much this parameter (extent of measures surface) can affect the results of the determinations. In Figure 1A, the machined specimen presents the grooves and small particles characteristic of the machining process. The smaller area (20 µm × 20 µm) shows the valleys and numerous depressions. The calculated rough mean square (RMS) coefficient was 0.30 µm and 0.12 µm, respectively (Table 2).

The surface topographies after the sand-blasting and acid-etching look very similar, independently of the grade of titanium. Both samples exhibit a granular structure with similar maximum heights of about 2 µm for the 50 × 50 area and about 1.5 µm for the 20 µm × 20 µm area. Both samples present numerous holes. The RMS coefficient was 0.46 µm and 0.38 µm for the 50 µm × 50 µm area, and 0.30 µm and 0.25 µm for the 20 µm × 20 µm area, as obtained for the sand-blasted/acid-etched grade 2 and the sand-blasted/acid-etched grade 4 surfaces, respectively (Table 2). The RMS coefficient was higher for the sand-blasted/acid-etched grade 2 than the machined and sand-blasted/acid-etched grade 4 surface. Hence, we concluded that the sand-blasted/acid-etched grade 2 modification had the highest roughness coefficient, and overall, the specimens had similar characteristics as the implants used in the first part of the study.

The AFM scanning results (Figure 2) showed higher roughness values for the sand-blasted/acid-etched surfaces, as compared with the machined ones.

### 3.2. Biological Studies

As the primary MSCs represent a rather heterogeneous cell population, the initial biological studies were carried out using pre-osteoblastic MC3T3-E1 cells, frequently used as an vitro model of osteogenesis. The MC3T3-E1 cells, derived from mouse calvaria, have provided a useful model for the analysis of the gene expression, as they are a non-transformed and relatively homogeneous cell line at a specific stage of differentiation, containing mostly pre-osteoblastic cells which can be induced to differentiate into mature osteoblasts. The cells were grown and maintained in vitro or induced to the osteogenic differentiation by treatment with specific osteogenic agents. As a first step, we evaluated the cell survival/proliferation of the pre-osteoblastic cell line seeded onto three different TiO_2_ surfaces, before and after the UVC irradiation. We compared the grade 2 titanium discs exposed to the machined or sand-blasted/acid-etched treatment with the grade 4 sand-blasted/acid-etched discs. The latter ones are more similar to the surface of the dental implants used clinically.

The cells were counted 24 h after seeding, as the initial attachment and proliferation of the cells is considered crucial, to achieve a successful osteointegration. The results, illustrated in Figure 3, show that in all discs the number of living cells after UV-photofunctionalization is highly significantly (*p* ˂ 0.001) increased, as compared to the untreated discs. As an example, Figure 4 shows two distinct fluorescent images of MC3T3 cells adhering to grade 4 sand-blasted/acid-etched titanium surface. The data, shown as a ratio between the living cells measured in the treated vs. non-treated disks, indicate a 2.5, 1.8 and 2.8 fold increase of the living cells in UVC-irradiated grade 2 machined discs, grade 2 sand-blasted/acid-etched discs and grade 4 sand-blasted/acid-etched discs, respectively. By the trypan blue exclusion, we also evaluated the percentage of the dead cells, which, following the UV-photofunctionalization, were virtually absent in the grade 2 machined disks, but equally present as viable cells in the irradiated grade 2 sand-blasted/acid-etched discs (Figure 3A). Figure 3B shows an example of the results obtained in grade 2 sand-blasted/acid-etched titanium discs, where an increased number of live cells was observed in the UV-photofunctionalized discs.

UV-photofunctionalization of the grade 4 sand-blasted/acid-etched discs resulted in a 2/3 reduction of dead cells, as compared to their frequency in the untreated discs (Figure 3C). The cells were also analyzed through fluorescence microscopy, after staining the cell nuclei with Hoechst dye.

As a second step, we moved to study the effects of the UVC treatment on the behavior of the primary MSCs, derived from the murine bone marrow, cultured either in expansion conditions or at different time intervals after the osteogenic induction. The choice of analyzing early times after the osteogenic treatment is based on previous studies by Picchi et al., where the osteogenic process has been thoroughly monitored, highlighting that the molecular events orchestrating the osteogenic commitment take place shortly after the osteogenic induction [22].

As in the experiments previously described on the pre-osteoblastic cell line, we first compared the percentage of the viable cells in the sand-blasted/acid-etched grade 4 titanium discs, before and after the UV-photofunctionalization. Figure 5 shows that, consistently with the data obtained using the MC3T3 cells, 24 h after seeding, the number of viable primary MSCs counted on the non-irradiated discs was half the number of cells grown onto the UV-photofunctionalized disks. Moreover, following the osteogenic induction, we observed a progressive and dramatic decrease of living cells in non-irradiated discs, as compared to UVC-treated disks. At 8 days of osteogenic differentiation, the cells present on the untreated disks were only a very small percentage (around 4%) of the cells grown on the irradiated surfaces.

Sand-blasted/acid-etched grade 4 titanium discs were next analysed for their ability to modulate in the MSC cells the expression of two osteogenic markers, Runx2 and ALP, at 3 and 8 days after the osteogenic induction. As previously pointed out, Runx2 is the master gene of osteogenesis, and Figure 6 shows that at 3 days of differentiation, its transcriptional activity is up-regulated (doubled) in the cells grown onto UV-photofunctionalized disks, as compared to the undifferentiated cells. However, this increase was not detected in the cells cultured onto non-irradiated disks. As a further control, the same analysis was carried out in the MSCs grown and differentiated in classical culture conditions, namely in plastic culture plates. Furthermore, control cells showed a 3-fold increase in the Runx2 expression after 3 days of the osteogenic differentiation. In addition, we observed that the transcriptional activity of the enzyme ALP is augmented in the cells grown on all surfaces.

Similar, and more clear-cut findings were observed after 8 days of the osteogenic induction, when the mRNA levels of both Runx2 and ALP were significantly increased only in cells seeded onto the UV-photofunctionalized discs (*p* ˂ 0.001, Figure 7). In contrast, virtually no variation was detected in the expression of both markers in MSCs grown onto non irradiated discs.

## 4. Discussion

A number of reports, focused on the biological effects of various surface modifications, have highlighted an inverse correlation between the proliferation and differentiation rates of the osteoblasts [23,24]. There is evidence showing that micro-roughened titanium surfaces have advantages over machined smooth surfaces in increasing both tissue-titanium mechanical interlocking and the osteoblastic differentiation [25], thus resulting in faster bone formation [26]. However, other studies have shown that the bone mass formed around rough surfaces is smaller than the one formed around machined ones [25], suggesting that rougher surfaces of material substrates may reduce the cell proliferation [20,27,28,29]. Therefore, it appeared that a surface modification sustaining both the osteoblast proliferation and differentiation was not available yet, and that only a compromise could be achieved. Here comes into play the importance of UV-photofunctionalization, as results have demonstrated that UVC-irradiation of both rough and smooth surfaces enhance the rate of both the osteoblast proliferation and differentiation.

In most studies, the cell attachment/proliferation on photofunctionalized surfaces was assessed at 3–24 h, using cell lines cultured on grade 2 TiO_2_. In all reports, a greater number of cells (on average a two-fold increase) was reported to adhere/grow onto the UVC-irradiated surfaces, as compared to the untreated surfaces. In our experiments, we have used a different strategy to assess the effect of UV-photofunctionalization. First, we have used not only the pre-osteoblastic cell line MC3T3, but also primary stem/precursor cells derived from murine bone marrow. In addition, we have compared the titanium surfaces pre-treated in different ways, before and after the UVC treatment. Second, we have compared the TiO_2_ disks of a different grade of purity. In our study we have analysed titanium grade 2, because it was previously utilized by other groups, and titanium grade 4, as in clinical practice, only grade 4 is used because of its mechanical properties. Furthermore, to make the study more reliable and standardized, the surfaces were not treated in the laboratory as in previous reports, but directly by the implantology company to simulate the clinical practice. Third, we monitored the cell survival/proliferation and differentiation not only at short times (24 h), but also up to 8 days of culture onto different surfaces. Our data, obtained the seeding of the pre-osteoblastic cell line onto grade 2 machined or sand-blasted/acid-etched, and grade 4 sand-blasted/acid-etched discs, demonstrate in all cases the positive effect of UV-photofunctionalization, resulting in a 2–3 fold increased cell attachment/proliferation after 24 h. Such an effect is in line with the previously reported observations [5,17]. Moreover, on the same discs, we also evaluated the percentage of the dead cells and observed different outcomes with the different types of discs. Following the UVC irradiation, there were virtually no dead cells on the grade 2 machined discs, while on grade 2 sand-blasted/acid-etched discs, the percentage of dead cells was similar as in non-irradiated discs, and on grade 4 sand-blasted/acid-etched discs, a 2/3 reduction was detected. Overall, the UV-photofunctionalization of the grade 4 sand-blasted/acid-etched surfaces, which are currently employed in dental implants, showed an encouraging effect for its clinical application. This conclusion is also supported by the fluorescence microscopy analysis (Figure 4) which showed a higher number of cells on the irradiated surfaces, as compared to the non-irradiated ones.

Furthermore, we chose to use in our study, primary MSC cells, which better mimic the situation in vivo, and these confirmed the beneficial effect of UV-photofunctionalization on the cell adhesion/growth after 24 h of culture, and provided evidence that such an effect is even stronger at later times (Figure 5). During 8 days of culture, the ratio between the viable cells present on the UVC-irradiated vs. non-irradiated surfaces, increased progressively, and, remarkably, at the end point of our analysis (8 days) a minority of cells (4%) were present on non-UV-photofunctionalized discs, as compared to the high percentage (96%) still detectable on the UVC treated discs. These results represent a significant extension and improvement of a previous observation by Aita et al., which was the only group to compare the growth of human MSC cells onto irradiated vs. non-irradiated surfaces. They reported that after 7 days of culture, the amount of cells present on the UVC-irradiated discs was 3–4 fold increased [30].

We also compared the osteogenic ability of MSC cells cultured on UVC-irradiated or non-irradiated grade 4 sand-blasted/acid-etched discs. The osteogenic differentiation was assessed by monitoring the gene expression of two key osteogenic markers, Runx2 and ALP, using the quantitative RT-PCR. Runx2 is a master gene of osteogenesis, as it plays a pivotal role in the commitment of multipotent mesenchymal cells to the osteoblastic lineage, and is required at early stages of the osteoblast differentiation. Moreover, it is able to up-regulate the expression of many bone matrix protein genes, including type 1 collagen, osteopontin, bone sialoprotein and osteocalcin. Thus, the analysis of the Runx2 expression is crucial to determine the onset of the molecular cascade of events that orchestrate the osteogenic differentiation. Previous studies have shown that days 3 and 8 after the osteogenic induction are optimal time points at which to prove that the osteogenic process is taking place [22]. Therefore, we compared the expression of both Runx2 and ALP in the MSCs seeded onto the UVC irradiated or non-irradiated discs, at day 3 and 8 after the osteogenic induction. ALP has been one of the first key players in the process of osteogenesis to be recognized. For this reason, it is a marker currently used to evaluate the osteogenic differentiation when assessing the phenotype or developmental maturity of the mineralized tissue cells. In the literature, a constant increase of the ALP expression has been reported after the UVC-irradiation of the titanium surfaces after 3, 7 and 10 days. Our results have confirmed the up-regulation of the ALP expression at both day 3 (cells grown on all surfaces, including the plastic culture plates, used as the internal controls) and day 8, particularly in the MSC cells seeded onto UV-photofunctionalized discs. Interestingly, the Runx2 expression has not been previously reported in the literature and our data provide the first evidence that its transcriptional activity is greatly enhanced in cells induced to differentiate onto UVC-irradiated surfaces. An increased expression (2 fold) at both time points is rather relevant, as it is known that changes in the gene expression of the transcription factors are rather limited, as compared to the variations occurring in the transcription of enzymes, such as ALP. Our results also suggest that Runx2 is a more reliable marker than ALP, which is known to be variable, mainly at early times of osteogenesis.

The improved survival/proliferation and differentiation of cells cultured onto UV-photofunctionalized discs may be accounted for by the observations reported by Iwasa et al. [31]. In this study, confocal microscopy images of osteoblasts after staining with rhodamine phalloidin showed that after 3 h of incubation, the cells seeded onto UV-treated titanium surfaces appeared definitely flatter and larger than the ones seeded on untreated surfaces. Moreover, the cells on UV treated titanium surfaces showed a clear stretch of lamellipodia-like actin projections, as well as cytoskeletons within their cytoplasm, whereas the majority of cells on the untreated surfaces were round shaped and did not exhibit the initiation of the elongating cell processes and developing cytoskeletons [31].

From a chemico-physical point of view, it could be speculated that the carboxyl groups present on protein structures become attracted by the titanium surface, thus resulting in a larger surface covered by cells, increasing the BIC and in turn creating a stronger osteointegration.

Our results suggest that the surfaces used in dental implantology can still be improved and that the UVC-irradiation of titanium enables an increase in osteoblastic proliferation without sacrificing the differentiation. Such a biological advantage was well shown by the higher cell numbers of the MSC cells detected after 8 days of culture on the UV-photofunctionalized disks, along with their increased expression of osteogenic markers. It can be speculated that such improved cell functions may be due to an improved interaction between the titanium surface and the cell adhesion proteins, as suggested in a previous study from our laboratories [21,32]. Elias et al. analyzed the relationship between the implant surface wettability and the cytokine production by blood cells [33]. In particular, on the hydrophobic surfaces, the presence was detected of antibodies that could reduce the cell adhesion. In contrast, both thrombins and prothrombins were predominant on the hydrophilic surfaces, and it is well known that these proteins play an important role in stimulating the cell adhesion to the biomaterial surface. In particular, it has been shown that thrombin may become conformationally altered in the post clotting wound environment, thus exposing the amino acid sequence (RGD) capable of interacting with the cell surface integrins, which would result in an increased ability of the cells to adhere to the photofunctionalized surface [34]. In turn, it has been reported that the integrin attachment has a direct role in modulating the expression of genes involved in both the cell proliferation and differentiation gene expression (1). Therefore, there is scientific evidence that after the carbon removal, the number of binding sites for proteins is increased, and can in turn improve the expression of the genes that control the cell proliferation and differentiation. Taken together, these findings might change the approach to the study of the implant surfaces and their modifications, by focusing on the biophysical interactions between the cell proteins and titanium surfaces.

Possible limitations of our present study lie in the fact that analysis of the TiO_2_ surfaces was restricted to the atomic force microscopy determinations, while additional important physical details would be obtained using scanning electron microscopy (SEM). This kind of investigation was however performed in our previous studies on UV-photofunctionalization, to which we are referring for further details [32,35]. Moreover, other genes besides Runx2 and ALP, are known to participate in the osteogenic differentiation, such as fibronectin 1, collagen 1A1, vinculin and matrix metalloproteases 2 and 9. The changes of expression of these latter genes following the exposure of cells to UV-irradiated TiO_2_ surfaces have indeed been investigated by us in a previous study, although using a different cellular model (human fibroblasts) [19].

## 5. Conclusions

UV-photofunctionalization is a new strategy in producing more reactive and biocompatible TiO_2_ surfaces, independently from the surface treatments performed during manufacturing. Overall, titanium surfaces after the UVC-irradiation can induce an increase in the cell attachment and proliferation and a decrease in the amount of dead cells, indicating that a virtually carbon-free surface is more biocompatible. Moreover, the increased cell proliferation positively correlated with the expression of the osteointegration master gene Runx2. As shown in previous studies from our and other laboratories, the UV-irradiation induces an effective removal of carbon-containing contaminants, resulting in an increased hydrophilicity, wettability and reactivity of TiO_2_ with O, N and S atoms present on proteins [32]. Such changes in the TiO_2_ surface chemical properties can largely explain the observed positive biological effects, and are liable to lead to the improved biocompatibility and integration of implants.

Altogether, our present study adds to and completes our previously reported observations, extending the investigation to the effects of UV-photofunctionalization on the additional osteointegration-related cell types and genes. As life expectancy is increasing, accompanied by systemic diseases which impair the cell metabolism, these studies should form a valid base for the clinical use of implant UV-photofunctionalization. By increasing the level and speed of osteointegration, UV-photofunctionalization can improve the quality of life in many patients being presently rehabilitated with removable partial or complete dentures when fixed solutions with implants are contraindicated.

## Figures and Tables

**Figure 1 jfb-13-00265-f001:**
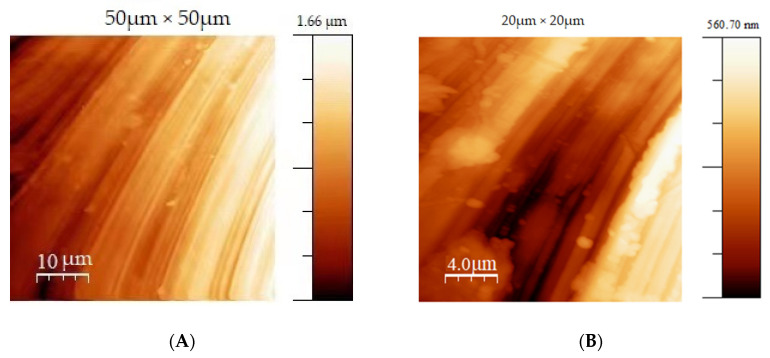
False color images resulting from the AFM analysis of the machined titanium discs for two different areas: (**A**) magnification 50 µm × 50 µm, (**B**) magnification 20 µm × 20 µm. Color intensities correspond to the distances in the z-direction, as shown in the respective scale bars.

**Figure 2 jfb-13-00265-f002:**
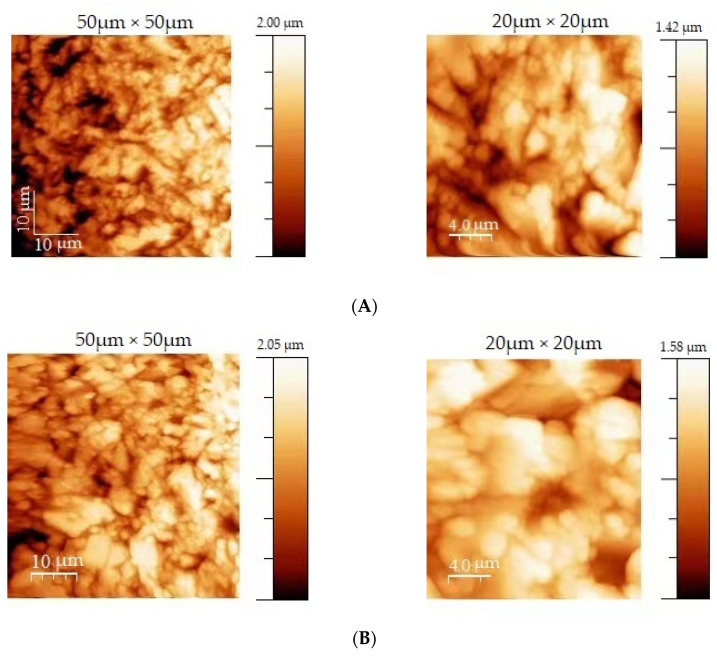
False color images resulting from the AFM analysis of the sand-blasted/acid-etched titanium specimens, performed on two different areas each: (**A**) Sand-blasted/acid-etched grade 2, (**B**) sand-blasted/acid-etched grade 4. Color intensities correspond to the distances in the z-direction, as shown in the respective scale bars.

**Figure 3 jfb-13-00265-f003:**
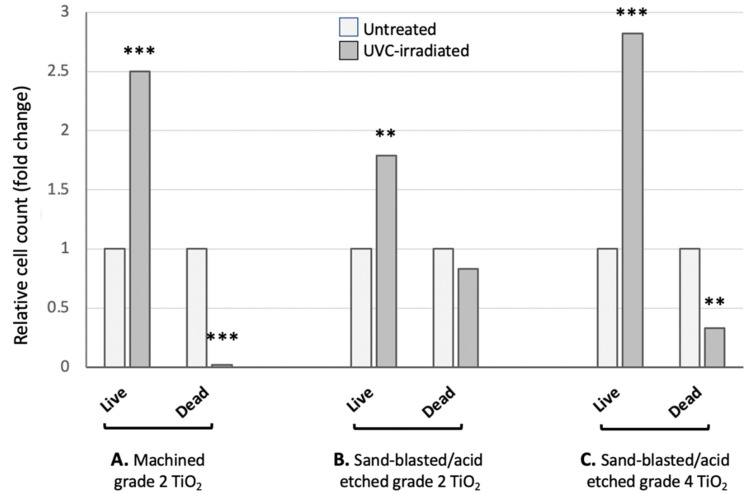
Attachment/proliferation MC3T3 after 24 h (**A**) results for the titanium grade 2 machined surface (**B**) results for the grade 2 sand-blasted/acid-etched titanium (**C**) results for the grade 4 sand-blasted/acid-etched titanium. Results shown are the means of two separate determinations. For the statistical analysis of the cell count obtained for the UVC-treated discs, it was compared to the non-treated ones, used as the control. ** *p* ˂ 0.01; *** *p* ˂ 0.001.

**Figure 4 jfb-13-00265-f004:**
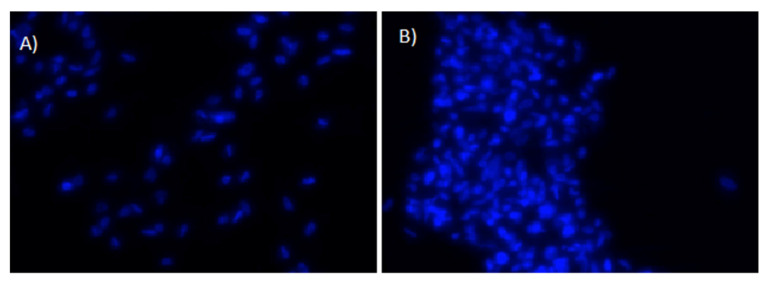
Fluorescence of MC3T3 cells nuclei stained with Hoechst dye, 24 h after seeding on a grade 4 sand-blasted/acid-etched titanium surface. Two distinct microscopic fields are shown (**A**,**B**); magnification: ×25.

**Figure 5 jfb-13-00265-f005:**
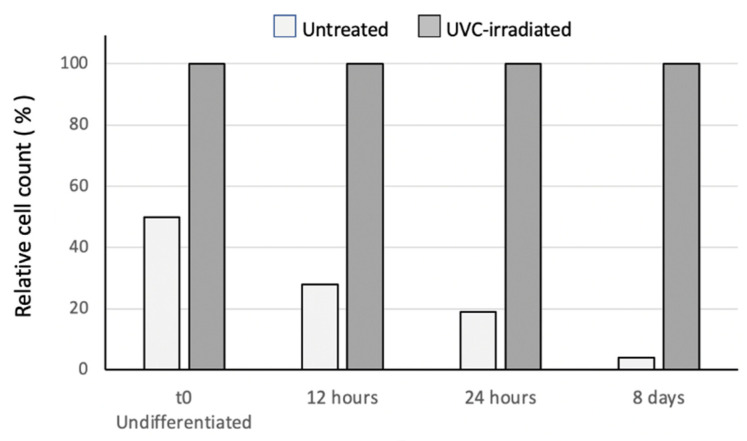
Cell attachment and proliferation of the MSC cells on the untreated vs. UVC-irradiated sand-blasted/acid-etched grade 4 titanium discs, at 12 h, 24 h and 8 days during their differentiation to the osteoblastic phenotype. Results shown are the means of two separate determinations.

**Figure 6 jfb-13-00265-f006:**
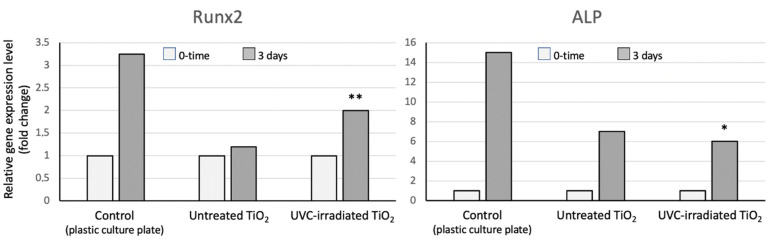
RT-PCR of the transcription factor Runx2 and enzyme alkaline phosphatase (ALP) in undifferentiated MSC cells and at 3 days after the proliferation/differentiation on the untreated vs. UVC-irradiated sand-blasted/acid-etched grade 4 titanium discs. Results shown are means of two separate determinations. For the statistical analysis, the RT-PCR results obtained for the photofunctionalized discs was compared to the non-treated and with no disc used as the control. * *p* ˂ 0.05; ** *p* ˂ 0.01.

**Figure 7 jfb-13-00265-f007:**
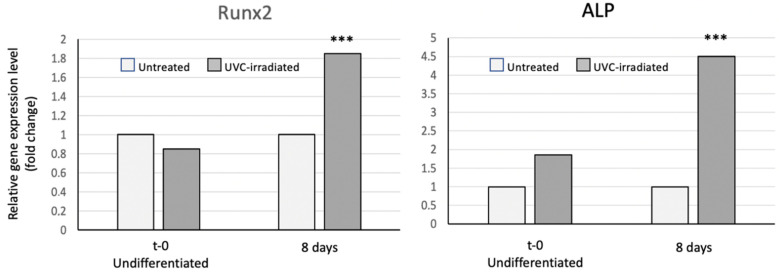
RT-PCR of the transcription factor Runx2 and alkaline phosphatase (ALP) in the undifferentiated MSC cells and after 8 days of the proliferation/differentiation on the untreated vs. UVC-irradiated sand-blasted/acid-etched grade 4 titanium discs. Results shown are the means of two separate determinations. RT-PCR results obtained with the UVC-treated discs were compared to the non-treated ones, used as the control. *** *p* ˂ 0.001.

**Table 1 jfb-13-00265-t001:** Primers and probes used in the real time PCR.

Gene Symbol	Gene Name	Primer Sequence (5′ > 3′)	Probe Sequence (5′ > 3′)
RUNX2	Runt-related transcription factor 2	F-TCTACCACCCCGCTGTCTTC R-TGGCAGTGTCATCATCTGAAATG	ACTGGGCTTCCTGCCATCACCGA
ALP	Alkaline Phosphatase	F-CCGTGGCAACTCTATCTTTGG R-CAGGCCCATTGCCATACAG	CCATGCTGAGTGACACAGACAAGAAGCC

**Table 2 jfb-13-00265-t002:** RMS coefficients for the different specimens in the different surface areas. For comparison, the values are also shown measured on a commercial sand-blasted acid-etched grade 4 TiO_2_ implant (Rapid™, Osteoplant Co., Poznan, Poland).

Area	Machined Surface (µm)	Sand-Blasted Acid-Etched Grade 2 (µm)	Sand-Blasted Acid-Etched Grade 4 (µm)	Dental Implant (µm)
50 × 50 µm	0.30	0.38	0.46	-
20 × 20 µm	0.12	0.25	0.30	-
15 × 15 µm	-	-	-	0.4

## Data Availability

The data presented in this study are available on request from the corresponding author.

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
