# Peer review of "Positive Effects of UV-Photofunctionalization of Titanium Oxide Surfaces on the Survival and Differentiation of Osteogenic Precursor Cells—An In Vitro Study"

_jfb, 2022, doi:10.3390/jfb13040265_

Round 1

Reviewer 1 Report

The manuscript represents the use of UV-photofunctionalization of titanium oxide surface on the attachment, survival, and differentiation of osteogenic precursor cells. The use of such a technique can indeed restore the biocompatibility of implant TiO2 surfaces. This is not a very recent technique as cited in the text. A recently updated literature survey can bring more interest to the introduction. The method part does not have the no. of replicated used, ANOVA if used and all the data regarding the statistical tools used. Also, grammatical mistakes need to be amended.

The manuscript can be recommended after a minor revision.

Author Response

  1. The manuscript represents the use of UV-photofunctionalization of titanium oxide surface on the attachment, survival, and differentiation of osteogenic precursor cells. The use of such a technique can indeed restore the biocompatibility of implant TiO2 surfaces. This is not a very recent technique as cited in the text. A recently updated literature survey can bring more interest to the introduction.

We agree that indeed UV-photofunctionalization has been the subject of a number of studies by now. The literature reviewed by us in our Introduction thus includes several early basic studies, but on the other hand we aldo discuss a few recent articles (published after 2014), which we believe have dealt with critical aspects of the matter relevant to our present investigation.

  1. The method part does not have the no. of replicated used, ANOVA if used and all the data regarding the statistical tools used.

The Methods section of our original submission actually does include a Statistics section (chapter 2.4.4., lines 162 ff. of revised ms). As a matter of fact, in view of the structure of our study and of the simple comparisons it required, methods applied were limited to the Student’s t test, setting p < 0.05 as the limit value for statistical significance.

  1. Also, grammatical mistakes need to be amended.

The text has been carefully reviewed and corrected where necessary.

Reviewer 2 Report

The novelty of the present study should be clearly presented in the introduction and results and discussion sections.

 The title is too long, I recommend shortening the title to make it attractive

The limitations of the study should be further emphasized.
LINE 43, "over over a period" check the grammar.

AFM image, please use English lang. to define the numbers inside image.

Atomic force microscopy (AFM) analysis is not enough, a more detailed SEM image should be added.

Author Response

  1. The novelty of the present study should be clearly presented in the introduction and results and discussion sections.

We thank the Reviewer for his useful suggestion. The Introduction, Discussion and Conclusions sections have been now integrated with passages highlighting the novelty of the findings presented in our manuscript.

  1. The title is too long, I recommend shortening the title to make it attractive

Following your suggestion we have modified the title to:

​Positive effects of UV-photofunctionalization of titanium oxide surfaces on survival and differentiation of osteogenic precursor cells - An in vitro study.

  1. The limitations of the study should be further emphasized.

We thank the Reviewer for his useful suggestion. The limitations of our study have been now emphasized in the text (end of the Discussion section).

  1. LINE 43, "over over a period" check the grammar. 

Thank you for noticing, the mistake has been corrected.

  1. AFM image LEGEND?, please use English lang. to define the numbers inside image. 

We apologize! The legends to Figs. 1 and 2 have been now integrated with the necessary information.

  1. Atomic force microscopy (AFM) analysis is not enough, a more detailed SEM image should be added.

We agree that SEM analysis would allow a more thorough appraisal of surface topography and the changes induced by UVC-irradiation. We have indeed performed SEM analyses in previous studies on UVC-photofunctionalized TiO2 [refs. 16,17,21]. We deemed however that AFM would anyway provide sufficient information for the pruposes of the present study, which was more “biological” in approach and first of all aimed at determining the effects of UVC-treatment on gene expression patterns in osteogenic precursors and mesenchymal cells.

Reviewer 3 Report

1. The title of the article needs to be revised, since the first part is sufficient for the purpose of the article, and the second part of the title clarifies the biological material.

2. The section "Abstract" of the article needs to be improved, as it contains redundant references to materials, methodology and conclusion.

3. What processing modes were used to obtain experimental titanium samples? What acid was used for the modification? They say that the method is patented, but where is the link to it?

4. What are 2 fragments with a surface area of 50*50 and 20*20 microns used for? Why do they have different roughness parameters? What explains this?

5. Why is an approximate root mean square (RMS) factor used and not the usual Ra or Rz? On what basis was this choice made?

6. AFM images are of poor quality and are presented at 2 magnifications, which are difficult to distinguish. It would be more useful to use scanning microscopy with similar magnifications of 20 or 50 µm.

7. Figures 1 and 2 use different languages and fonts.

8. In the materials science part of the article, there are no results of changes in the properties of materials as a result of UV photofunctionalization. There are only reliable results for changes in attachment/proliferation, RT-PCR transcription factor Runx2 and alkaline phosphatase (AP). But it talks about "changing the chemical structure of the surface." It is necessary to prove that they are of a chemical nature.

Author Response

  1. The title of the article needs to be revised, since the first part is sufficient for the purpose of the article, and the second part of the title clarifies the biological material.

Following your suggestion we have modified the title to:

​Positive effects of UV-photofunctionalization of titanium oxide surfaces on survival and differentiation of osteogenic precursor cells - An in vitro study.

  1. The section "Abstract" of the article needs to be improved, as it contains redundant references to materials, methodology and conclusion.

Thank you for this remark. The Abstract has been entirely reformulated as suggested.

  1. What processing modes were used to obtain experimental titanium samples? What acid was used for the modification? They say that the method is patented, but where is the link to it?

Titanium samples were etched first with hydrofluoric acid at room temperature, followed by a further etching step with a mixture containing sulfuric acid. All disks were then rinsed in water and plasma cleaned with argon to remove any potential by-products. The procedure was also used in previous work from our laboratory [n. 35]. We have now checked that this procedure is actually not included in patents owned by our industrial partner (Osteoplant Co., Poznan, PL). The text has been integrated and modified accordingly.

  1. What are 2 fragments with a surface area of 50x50 and 20x20 microns used for? Why do they have different roughness parameters? What explains this?

Thank you for your question. The two different area sizes shown were chosen just to evaluate how this can parameter can affect results of analysis. This has been now explained in the text (Methods section). On the micron scale, the surfaces considered in fact appear so rough that moving from one region of the sample to another can likely produce different parameters. The general decrease in the roughness parameter observed in transitioning from a larger area (50x50) to a smaller one (20x20) can in principle be attributed to statistical reasons. Holes and peaks of a surface actually have finite widths, and thus, if the size of surface measurements is progressively reduced, determinations would eventually describe the flat bottom of a groove or the plateau on a hill, wherefor RMS values might end up being zero.

  1. Why is an approximate root mean square (RMS) factor used and not the usual Ra or Rz? On what basis was this choice made?

Thank you for your question. In our opinion, results obtainable with different calculations can be considered as largely equivalent. The root mean square (RMS, also called Rq) parameter employed in the present study is actually used in the literature as often as Ra and Rz factors. The difference among these values lies in the procedures used to analyze the profile of heights of the surface measured at several points. In the case of RMS, over a series of measured points the average is obtained of squared differences between heights measured at individual points and the average height of the profile, then calculating the square root of this average. In the case of Ra, a mean value is obtained using absolute differences between heights at individual points and the average height. In turn, Rz is calculated as the average of differences between maximum and minimum heights. The numerical differences between RMS and Ra values are of minor magnitude, while slightly larger differences can be observed with Rz. When different surfaces are compared as to their relative roughnesses, both RMS and Ra usually give similar evaluations. RMS is the factor routinely used in our laboratories, and based on the experience we have previously accumulated we decided to use RMS also for the present study.

  1. AFM images are of poor quality and are presented at 2 magnifications, which are difficult to distinguish. It would be more useful to use scanning microscopy with similar magnifications of 20 or 50 µm.

We agree that SEM analysis would allow a more thorough appraisal of surface topography and the changes induced by UVC-irradiation. We have indeed performed SEM analyses in previous studies on UVC-photofunctionalization [refs. 16,17,21]. We deemed however that AFM would anyway provide sufficient information for the pruposes of the present study, which was more “biological” in its approach and first of all aimed at determining the effects of UVC-treatment on the gene expression patterns in osteogenic precursors and mesenchymal cells.

  1. Figures 1 and 2 use different languages and fonts.

Thank you for noticing, the legends have been corrected with a uniform style.

  1. In the materials science part of the article, there are no results of changes in the properties of materials as a result of UV photofunctionalization. There are only reliable results for changes in attachment/proliferation, RT-PCR transcription factor Runx2 and alkaline phosphatase (AP). But it talks about "changing the chemical structure of the surface." It is necessary to prove that they are of a chemical nature.

Chemical changes in the surface of UVC-irradiated TiO2 mostly consist in increased hydrophilicity and wettability, and were actually analyzed and discussed in detail in previous studies from our laboratory [refs. 30,31]. We apologize for not having made this point enough clear. The text has been now integrated with additional details, recalling the references concerned (Conclusions section, lines 434 ff.).